# TopoMLP: A Simple yet Strong Pipeline for Driving Topology Reasoning

**Dongming Wu**[1]⋆**, Jiahao Chang**[2]⋆**, Fan Jia**[3]**, Yingfei Liu**[3]**, Tiancai Wang**[3]‡**, Jianbing Shen**[4]†

[1] Beijing Institute of Technology, [2] University of Science and Technology of China,
[3] MEGVII Technology, [4] SKL-IOTSC, University of Macau
{wudongming97, shenjianbingcg}@gmail.com, wangtiancai@megvii.com

## Abstract

Topology reasoning aims to comprehensively understand road scenes and present drivable routes in autonomous driving. It requires detecting road centerlines (lane) and traffic elements, further reasoning their topology relationship, *i.e.*, lane-lane topology, and lane-traffic topology. In this work, we first present that the topology score relies heavily on detection performance on lane and traffic elements. Therefore, we introduce a powerful 3D lane detector and an improved 2D traffic element detector to extend the upper limit of topology performance. Further, we propose TopoMLP, a simple yet high-performance pipeline for driving topology reasoning. Based on the impressive detection performance, we develop two simple MLP-based heads for topology generation. TopoMLP achieves state-of-the-art performance on OpenLane-V2 dataset, *i.e.*, 41.2% OLS with ResNet-50 backbone. It is also the 1st solution for 1st OpenLane Topology in Autonomous Driving Challenge. We hope such simple and strong pipeline can provide some new insights to the community. Code is at https://github.com/wudongming97/TopoMLP.

## 1 Introduction

Understanding the topology in road scenes is an important task for autonomous driving, since it provides the information about the drivable region as well as the traffic signal. Recently, the topology reasoning task has raised great attention in the community thanks to its crucial application in ego planning (Chai et al., 2020; Casas et al., 2021; Hu et al., 2023). In specific, given multi-view images, topology reasoning aims to learn vectorized road graphs between the centerlines and traffic elements (Li et al., 2023; Wang et al., 2023). It consists of four primary tasks, centerline detection, traffic element detection, lane-lane topology, and lane-traffic topology reasoning.

Different from the conventional perception pipelines that include multiple independent tasks (Li et al., 2022b; Liu et al., 2023b), these four tasks naturally have a logical order, *i.e.*, first-detect-then-reason. If some lane and traffic instances are not detected, the corresponding topology connection will be missed, as illustrated in the right of Fig. 1. It naturally leads to a question: *What is the extent of the quantitative effect of basic detection on topology reasoning?* To answer this question, we conduct detailed ablation studies on detection performance by varying the backbones. It shows that the topology performances are constantly improved with stronger detection. When the basic detection is frozen, we find that replacing the topology prediction with the ground truth (GT) introduces minor improvements. For example, when using Swin-B backbone, the $TOP_{ll}$ and $TOP_{lt}$ scores with topology GT are 10.0% and 30.9%, which are only higher than using topology prediction by 0.5% and 2.6%, respectively. This phenomenon encourages us to prioritize the design of two detectors.

In specific, we employ two query-based detection branches: one (Liu et al., 2023b) dedicated to the detection of 3D centerlines, and another one (Zhu et al., 2021) for 2D traffic detection. The 3D lane detector utilizes a smooth lane representation and interprets each lane query as a set of control points of a Bézier curve. Inspired by MOTRv2 (Zhang et al., 2023), the performance of 2D traffic

---

⋆This work was done during internship at MEGVII. ‡Project leader. †Corresponding author: Jianbing Shen.
    This work was supported in part by the FDCT grants 0102/2023/RIA2, 0154/2022/A3, and 001/2024/SKL, the MYRG-CRG2022-00013-IOTSC-ICI grant and the SRG2022-00023-IOTSC grant.

| Backbone | DET$_l$ | DET$_t$ | Topo GT | TOP$_{ll}$ | TOP$_{lt}$ | OLS |
|---|---|---|---|---|---|---|
| ResNet-50 | 28.3 | 50.0 | ✓ | 7.2
7.5 (+0.3) | 22.8
24.4 (+1.6) | 38.2
38.3 (+0.1) |
| VOV | 29.7 | 52.1 | ✓ | 7.9
8.5 (+0.6) | 25.6
27.5 (+1.9) | 40.1
40.8 (+0.9) |
| Swin-B | 30.7 | 54.3 | ✓ | 9.5
10.0 (+0.5) | 28.3
30.9 (+2.6) | 42.2
43.1 (+0.9) |

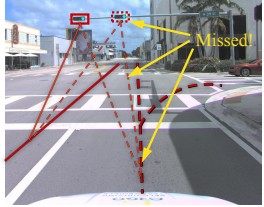

Figure 1: **Left**: The evaluation of topology when using ground-truth topology (represented as "Topo GT") to replace predicted topology while retaining the detection results. **Right**: The illustration of how missing detections (represented by dashed lines) influence topology reasoning. They uncover an essential truth: *the fundamental detections are paramount to the topology reasoning*.

detector can be further enhanced by adding an additional YOLOv8 (optional) object detector thanks to its advantage on detecting small objects, such as traffic lights.

Despite the basic detection, another challenge in driving topology reasoning is how to effectively model the connection between lanes and traffic elements. Prior works (Langenberg et al., 2019; Can et al., 2021; 2022) employ a straightforward method, which uses a multi-layer perceptron (MLP) to predict the topology relationship. However, they mainly focus on associating different lanes in image domain. To cope with the 3D space, some follow-up methods (Li et al., 2023; Xu et al., 2023) tend to utilize graph-based modeling to predict topology structure.

In this paper, we develop a simple yet effective framework, termed TopoMLP, for topology reasoning. Our work is inspired by the pairwise representation in human-object interaction detection (Gao et al., 2018; Chao et al., 2018; Wang et al., 2019), similar to topology reasoning. The pairwise representation is constructed by encoding the human/object pair boxes into two mask embeddings. These embeddings are concatenated together and further used to perform action classification by a simple MLP. We wonder if it is possible to develop a simple MLP-based framework for sufficiently understanding the relationships in driving topology reasoning. Taking lane-lane topology as an example, if the lanes are accurately predicted, the intersection points (see Fig. 2) between lanes can be easily reasoned to be overlapped. As for the lane-traffic topology, the traffic elements can be easily matched with the corresponding centerlines by the relative location between traffic bounding boxes and lane points. Therefore, a simple MLP seems enough for efficient topology reasoning. Specifically, we convert the query representations of both traffic elements and centerlines into two embeddings and concatenate them together for topology classification by an appended MLP.

Moreover, we notice that the topology metrics of OpenLane-V2 have some drawbacks. It uses graph-based mAP, while it focuses more on the order of predictions. Some false positives from unmatched lanes or traffic elements are defaulted to a high confidence score, *i.e.*, 1.0. Accordingly, manually decreasing the priority of these false positive predictions (or increasing the priority of true positive predictions) enables to improve the overall mAP score by a large margin. To tackle this problem, we suggest to include a correctness factor based on existing topology metric to correct the drawback.

Our contributions are summarized as four-fold. **First**, we provide an in-depth analysis of the nature of driving topology reasoning. It requires following a "first-detect-then-reason" philosophy for better topology prediction. **Second**, we propose a simple but strong model, named TopoMLP. It includes two well-designed high-performance detectors and two elegant MLP networks with position embedding for topology reasoning. **Third**, we claim that the current topology reasoning evaluation possesses a significant loophole. To rectify this, we enhance the topology metric by incorporating a correctness factor. **Fourth**, all experiments are conducted on the popular driving topology reasoning benchmark, OpenLane-V2, showing TopoMLP reaches state-of-the-art performance. Besides, TopoMLP ranks 1st of 1st OpenLane Topology in Autonomous Driving Challenge (Wu et al., 2023).

## 2   RELATED WORKS

### 2.1   LANE DETECTION METHOD

For a long time, detecting lane markings has been one of the most important topics in autonomous driving. Prior works usually use appearance and geometric cues to detect the road (Tan et al., 2006;

Alvarez & López, 2010; Paz et al., 2015). With the advancement of deep learning, the development of lane detection has made great progress. Among them, some methods attempt to use a segmentation map to describe road lane (Batra et al., 2019; Can et al., 2022; He & Balakrishnan, 2022). Currently, vector-based methods have become mainstream because they can deal well with 3D lane detection (Garnett et al., 2019; Guo et al., 2020; Yan et al., 2022; Chen et al., 2022). However, these methods base a set of predefined Y-axis points in the query to predict 3D lanes, which fail to make the 3D lane prediction only across the Y axis. More recently, TopoNet (Li et al., 2023) models each lane into an anchor query, but it misses the lane prior with a smoothed curve. *In our study, we make full use of this prior to providing a smoother representation.*

## 2.2 LANE TOPOLOGY LEARNING

Learning lane topology plays an important role in scene understanding for autonomous driving. Earlier works (Chu et al., 2019; Homayounfar et al., 2019; He et al., 2020; Bandara et al., 2022) focus on generating road graphs from aerial images. However, using aerial images is unreasonable for ongoing vehicles. Therefore, directly using vehicle-mounted sensors to detect lane topology has become popular due to their valuable application. STSU (Can et al., 2021) uses a Transformer-based model to detect centerlines and objects together, and then predict centerline association formatted to a directed graph by an MLP. TopoRoad (Can et al., 2022) further introduces additional minimal cycle queries to ensure the preservation of the order of intersections. Can et al. (Can et al., 2023) also provide additional supervision of the relationship by considering the centerlines as cluster centers to assign objects and greatly improve the lane graph estimation. LaneGAP (Liao et al., 2023) designs a heuristic-based algorithm to recover the graph from a set of lanes. CenterLineDet (Xu et al., 2023) and TopoNet (Li et al., 2023) regard centerlines as vertices and design a graph model to update centerline topology. *In this work, we focus on lane topology nature and employ a simple and elegant position embedding to enhance topology modeling.*

## 2.3 HD MAP PERCEPTION

HD Map Perception aims to comprehend the layout of the driving scene, such as lanelines, pedestrian crossing, and drivable areas, mirroring the concept of driving scene reasoning. The recent research focuses on learning HD maps using segmentation and vectorization techniques to meet low-cost requirements. HDMapNet (Li et al., 2022a) explores grouping and vectorizing the segmented map with complicated post-processings. VectorMapNet (Liu et al., 2023a) directly uses a sequence of points to represent each map element, further decoding laneline locations. Some follow-up methods propose different modeling strategies to represent the sequence of points, such as the permutation-based (Liao et al., 2022), the piecewise Bézier curve (Qiao et al., 2023), the pivot-based map (Ding et al., 2023). Different from the aforementioned approaches, our method employs simple and elegant modeling, each query referring to a lane.

## 3 METHOD

In this section, we elaborate on TopoMLP, a unified query-based framework for driving topology reasoning. It is able to effectively accomplish four different tasks in a single framework, including lane detection, traffic element detection, lane-lane topology, and lane-traffic topology prediction. The overall pipeline of TopoMLP is shown in Fig. 2. More details are described as follows.

### 3.1 LANE DETECTOR

Our lane detector is inspired by the advanced 3D multi-view object detector PETR (Liu et al., 2022; 2023b), which first introduces 3D position embedding (3D PE) into the query-based framework DETR (Carion et al., 2020; Zhu et al., 2021). In this work, we represent each centerline as a smooth Bézier curve with $M$ control points within 3D space and each curve refers to a lane query. Our lane detector performs direct interaction between lane queries with multi-view visual features in transformer decoder and outputs control points, further transformed to lane coordinates.

Formally, given multi-view images from camera sensors, we first employ a backbone (*e.g.*, ResNet-50 (He et al., 2016)) to generate feature maps $\boldsymbol{F} \in \mathbb{R}^{V \times C \times H \times W}$, where $V$, $C$, $H$, and $W$ represent

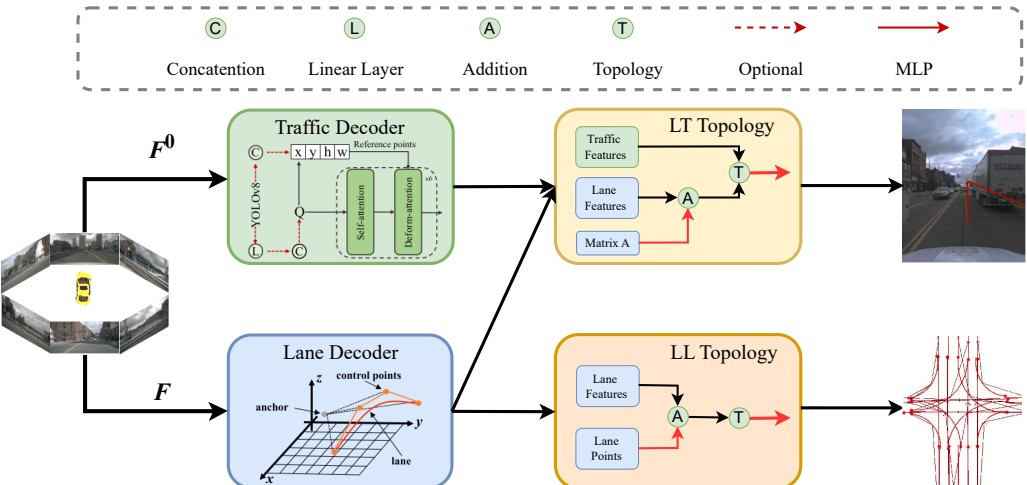

Figure 2: **The overall architecture of TopoMLP.** The lane decoder depicts each centerline as a Bézier curve for a smooth representation. The traffic decoder is optionally enhanced by additional YOLOv8 proposals. The prediction of lane-traffic (LT) and lane-lane (LL) topology is accomplished by an MLP with position embedding. "Topology" means an operation in § 3.3.

the view number, channel, height, and width of the features, respectively. The 3D PE is encoded into the visual features to generate position-aware features following (Liu et al., 2022). Then we initialize $N_L$ learnable 3D lane anchor points, denoted as $Q^L \in \mathbb{R}^{N_L \times 3}$. After projecting the feature dimension of anchor points from 3 to $C$ using a position encoding and a linear layer, we further feed it into transformer decoder to update the lane query features $\hat{Q}^L$:

$$\hat{Q}^L = \textbf{LaneDecoder}(F, \textbf{Linear}(Q^L)) \in \mathbb{R}^{N_L \times C}. \tag{1}$$

where the LaneDecoder is a stack of Transformer decoder layers. On top of the transformer decoder, we adopt two independent MLPs to predict the offset of control points and the classification scores, respectively. The final control point outputs are ordered and obtained by adding basic anchor points with the relative offsets. The control points are transformed into lane points for training and testing.

## 3.2    TRAFFIC ELEMENT DETECTOR

The prevalent approaches for traffic element detection in driving topology reasoning are mainly query-based and end-to-end deployed (Li et al., 2023; Kalfaoglu et al., 2023; Lu et al., 2023). Although such straightforward end-to-end implementation is appealing, the detection performance is much inferior to the specialized 2D detectors, such as YOLO series, due to small objects and class imbalance problems. To address these limitations, we propose to *optionally* improve the query-based detectors by elegantly incorporating an extra object detector YOLOv8.

Our traffic element detector typically follows the head design in Deformable DETR (Zhu et al., 2021) to predict bounding boxes and classification scores. It adopts query embeddings to generate a set of reference points as anchors. We modify the reference format into reference boxes with the center points, height, and width. As an alternative, the high-quality proposals from YOLOv8 can serve as an anchor box initialization, providing better local priors. It greatly eases the trade-off between topology reasoning and traffic detection.

Specifically, we first collect the multi-scale feature maps of the front view from multi-view features $F$, denoted as $F^0$. YOLOv8 takes $F^0$ as input and generates multiple proposals, which are concatenated with a set of reference boxes produced from randomized queries, denoted as $R^T$. The generated boxes by YOLOv8 are encoded by sine-cosine embedding to generate query features, which are concatenated with the randomized queries, denoted as $Q^T$. The query features as well as the reference boxes are fed into the deformable decoder:

$$\hat{Q}^T = \textbf{TrafficDecoder}(F^0, Q^T, R^T) \in \mathbb{R}^{N_T \times C}, \tag{2}$$

where the TrafficDecoder is a stack of Deformable decoder layers. Based on the decoded traffic features $\hat{Q}^T$, we implement two independent MLPs for bounding box classification and regression.

### 3.3 LANE-LANE TOPOLOGY REASONING

Lane-lane topology reasoning branch aims to predict the lane-lane connection relationship. To incorporate the discriminative lane information, we integrate the predicted lane points into the lane query features. In specific, we implement MLP to embed the lane coordinates and then add them into the decoded lane query features $\hat{\boldsymbol{Q}}^L \in \mathbb{R}^{N_L \times C}$. For notion simplicity, we still use $\hat{\boldsymbol{Q}}^L$ to represent the integrated query features. They are repeated $N_L$ times, generating two features with sizes $N_L \times (N_L) \times C$ and $(N_L) \times N_L \times C$, where $(N_L)$ defines different repeating directions. After concatenation operation generating $\hat{\boldsymbol{Q}}^{LL} \in \mathbb{R}^{N_L \times N_L \times 2C}$, we apply MLP to perform binary classification:

$$\boldsymbol{G}^{LL} = \textbf{MLP}(\hat{\boldsymbol{Q}}^{LL}) \in \mathbb{R}^{N_L \times N_L}, \tag{3}$$

where the $\boldsymbol{G}^{LL}$ is lane-lane topology prediction.

### 3.4 LANE-TRAFFIC TOPOLOGY REASONING

The key idea of our lane-traffic topology reasoning is to project two kinds of features into the same space. Given the lane query embedding $\hat{\boldsymbol{Q}}^L \in \mathbb{R}^{N_L \times C}$ from 3D space, we sum the view transformation matrix $\boldsymbol{A} \in \mathbb{R}^{3 \times 3}$ from 3D to perspective view with it, *i.e.*, $\hat{\boldsymbol{Q}}^L + \text{MLP}(\boldsymbol{A})$. Here, the view transformation matrix $\boldsymbol{A}$ is formulated in terms of camera intrinsic and extrinsic. Similar to lane-lane topology, the transformed lane query features and the traffic query embedding $\hat{\boldsymbol{Q}}^T \in \mathbb{R}^{N_T \times C}$ are transformed into $\hat{\boldsymbol{Q}}^{LT} \in \mathbb{R}^{N_L \times N_T \times 2C}$ through repeating and concatenating operations. An MLP network is used to generate lane-traffic topology prediction $\boldsymbol{G}^{LT}$:

$$\boldsymbol{G}^{LT} = \textbf{MLP}(\hat{\boldsymbol{Q}}^{LT}) \in \mathbb{R}^{N_L \times N_T}. \tag{4}$$

### 3.5 LOSS FUNCTION

Our final loss function is defined as follows:

$$\mathcal{L} = \mathcal{L}_{det_l} + \mathcal{L}_{det_t} + \mathcal{L}_{top_{ll}} + \mathcal{L}_{top_{lt}}, \tag{5}$$

where $\mathcal{L}_{det_l}$ is lane detection loss, which includes a focal loss (Lin et al., 2017) supervising classification and an L1 loss for lane regression. $\mathcal{L}_{det_t}$ is traffic element detection loss, which has a focal loss for classification, a $\mathcal{L}_1$ loss and a GIoU loss for bounding box regression. The lane-lane topology loss $\mathcal{L}_{top_{ll}}$ contains a focal loss for binary classification and an L1 loss between the matched lane points in terms of the topology ground-truth. The lane-traffic topology loss $\mathcal{L}_{top_{lt}}$ is a focal loss for binary classification. Since our TopoMLP is a query-based method, it requires the matching between the predictions and ground-truth. In this work, we only use bipartite matching on the basic lane and traffic element detection. The matching is directly used in topology reasoning loss as well.

## 4 EXPERIMENTS

### 4.1 DATASET AND METRIC

**Dataset.** The experiments are conducted on the OpenLane-V2 (Wang et al., 2023). OpenLane-V2 is a large-scale perception and reasoning dataset for scene structure in autonomous driving. It has two subsets, *i.e.* $subset\_A$ and $subset\_B$, developed from Argoverse 2 (Wilson et al., 2021) and nuScenes (Caesar et al., 2020), respectively. Each subset comprises 1,000 scenes with annotations at $2Hz$. Note that the $subset\_A$ contains seven views and $subset\_B$ contains six views.

**Evaluation Metric.** These two basic detections require measuring instance-level performance. Therefore, the perception metrics, including $\text{DET}_l$ and $\text{DET}_t$, are mean average precision (mAP) following the work (Wang et al., 2023). Specifically, $\text{DET}_l$ employs the Fréchet distance for quantifying similarity and is averaged over match thresholds set at $\{1.0, 2.0, 3.0\}$. On the other hand, $\text{DET}_t$ employs Intersection over Union (IoU) as the similarity measure, with averages calculated over various traffic categories. For topology metrics, the TOP score also employs an mAP metric, which is designed specifically for graph data. To summarize the overall effect of primary detection and topology reasoning, the OpenLane-V2 Score (OLS) is conducted as:

$$\text{OLS} = \frac{1}{4}[\text{DET}_l + \text{DET}_t + f(\text{TOP}_{ll}) + f(\text{TOP}_{lt})], \tag{6}$$

where $f$ is the square root function.

| Method | Backbone | Epoch | $DET_l$ | $DET_t$ | $TOP_{ll}$ | $TOP_{lt}$ | OLS |
|---|---|---|---|---|---|---|---|
| STSU (Can et al., 2021) | ResNet-50 | 24 | 12.7 | 43.0 | 0.5 | 15.1 | 25.4 |
| VectorMapNet (Liu et al., 2023a) | ResNet-50 | 24 | 11.1 | 41.7 | 0.4 | 5.9 | 20.8 |
| MapTR (Liao et al., 2022) | ResNet-50 | 24 | 17.7 | 43.5 | 1.1 | 10.4 | 26.0 |
| TopoNet (Li et al., 2023) | ResNet-50 | 24 | 28.5 | 48.1 | 4.1 | 20.8 | 35.6 |
| TopoMLP | ResNet-50 | 24 | 28.3 | 50.0 | 7.2 | 22.8 | 38.2 |
| TopoMLP* | ResNet-50 | 24 | **28.8** | **53.3** | **7.8** | **30.1** | **41.2** |
| TopoMLP | VOV | 24 | 29.7 | 52.1 | 7.9 | 25.6 | 40.1 |
| TopoMLP | Swin-B | 24 | 30.7 | 54.3 | 9.5 | 28.3 | 42.2 |
| TopoMLP* | Swin-B | 24 | 30.0 | 55.8 | 9.4 | 31.7 | 43.3 |
| TopoMLP | Swin-B | 48 | 32.5 | 53.5 | 11.9 | 29.4 | 43.7 |

Table 1: **Performance comparison with state-of-the-art methods** on OpenLane-V2 $subset\_A$ set. Results for existing methods are from TopoNet. TopoMLP is trained end-to-end, while '*' indicates using extra YOLOv8 proposals. The best is in bold.

| Method | Backbone | Epoch | $DET_l$ | $DET_t$ | $TOP_{ll}$ | $TOP_{lt}$ | OLS |
|---|---|---|---|---|---|---|---|
| STSU (Can et al., 2021) | ResNet-50 | 24 | 8.2 | 43.9 | 0.0 | 9.4 | 21.2 |
| VectorMapNet (Liu et al., 2023a) | ResNet-50 | 24 | 3.5 | 49.1 | 0.0 | 1.4 | 16.3 |
| MapTR (Liao et al., 2022) | ResNet-50 | 24 | 15.2 | 54.0 | 0.5 | 6.1 | 25.2 |
| TopoNet (Li et al., 2023) | ResNet-50 | 24 | 24.3 | 55.0 | 2.5 | 14.2 | 33.2 |
| TopoMLP | ResNet-50 | 24 | **26.6** | **58.3** | **7.6** | **17.8** | **38.7** |
| TopoMLP | VOV | 24 | 29.6 | 62.2 | 8.9 | 20.5 | 41.7 |
| TopoMLP | Swin-B | 24 | 32.3 | 65.5 | 10.5 | 23.2 | 44.6 |

Table 2: **Performance comparison with state-of-the-art methods** on OpenLane-V2 $subset\_B$ set. Results for existing methods are from TopoNet.

## 4.2 IMPLEMENTATION DETAILS

**Feature Extractor.** All images are resized into the same resolution of $1550 \times 2048$, and are downsampled with a ratio of 0.5. We implement different backbones, *i.e.*, ResNet-50 (He et al., 2016), VOV (Lee et al., 2019), and Swin-B (Liu et al., 2021) for feature extraction. The number of output channels is set to $C = 256$. For lane detection, the C5 feature is upsampled and fused with C4 feature using FPN. For traffic detection, the C3, C4, and C5 features are used as the feature pyramid.

**Lane Detector.** The lane query number is set to $N_L = 300$, and the number of control points is 4. During training, the control points are transformed into 11 lane points for calculating loss. We set the region to $[-51.2m, 51.2m]$ on the X-axis, $[-25.6m, 25.6m]$ on the Y-axis, and $[-8m, 4m]$ on the Z-axis. The lane detection head is composed of 6 transformer decoder layers. The MLP heads contain two fully connected layers with ReLU activation. For lane detection loss $\mathcal{L}_{det_l}$, the weight of the classification part is 1.5, and the weight of the regression part is 0.2.

**Traffic Detector.** The decoder architecture follows the original designs of Deformable DETR (Zhu et al., 2021). The number of random queries in the traffic decoder is 100. The detection results from YOLOv8[1] are stored in advance. The weight of the classification loss is 1.0, the weight of L1 loss is 2.5, and the weight of GIoU loss is 1.0.

**Topology Head.** The MLP network used in two topology heads consists of three linear layers with ReLU activation. We represent the lane-lane topology loss with L1 loss and classification loss as $\mathcal{L}_{top_{ll}} = \lambda_{L1}\mathcal{L}_{L1} + \lambda_{cls}\mathcal{L}_{cls}$, where $\lambda_{L1} = 0.1$ and $\lambda_{cls} = 5$. The loss coefficient of lane-traffic topology loss $\mathcal{L}_{top_{lt}}$ is 0.5.

**Training and Inference.** The overall model TopoMLP is trained by AdamW optimizer (Loshchilov & Hutter, 2017) with a weight decay of 0.01. The learning rate is initialized with $2.0 \times 10^{-4}$ and decayed with cosine annealing policy (Loshchilov & Hutter, 2016). We adopt the HSV augmentation and grid mask strategy for training. All the experiments are trained for 24 epochs on 8 Tesla A100

---

[1]The official codes we adopt are available at https://github.com/ultralytics/ultralytics.

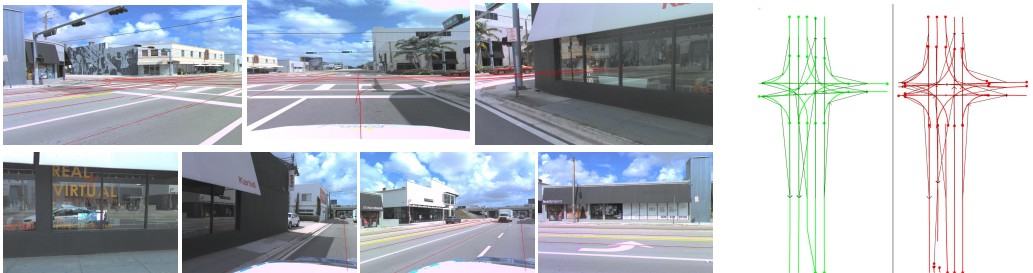

Figure 3: **Qualitative results for lane detection and lane-lane topology of TopoMLP.** Given the multi-view images, our method can predict the most lanes and connect them correctly under various challenges, like occluded lanes and complicated intersections. The green lanes are ground-truth and the red lanes are predictions, which are projected into images and BEV map.

GPUs with a batch size of 8 if not specified. During the inference time, our model outputs at most 300 lanes for evaluation. Other post-processing techniques are not implemented.

### 4.3 STATE-OF-THE-ART COMPARISON

We compare TopoMLP with the state-of-the-art approaches, such as STSU (Can et al., 2021), VectorMapNet (Liu et al., 2023a), MapTR (Liao et al., 2022), TopoNet (Li et al., 2023). Table 1 shows the results on $subset\_A$ of OpenLane-V2. Without bells and whistles, our method achieves 38.2 OLS using ResNet-50 backbone, surpassing other state-of-the-art methods. Compared to TopoNet, our approach shows a much better topology reasoning accuracy (7.2 *v.s.* 4.1 on $TOP_{ll}$, 22.8 *v.s.* 20.8 on $TOP_{lt}$) while also achieves decent detection accuracy (28.3 *v.s.* 28.5 on $DET_l$, 50.0 *v.s.* 48.1 on $DET_t$). For a better performance, we apply a more powerful backbone and more training time: when using Swin-B for training 48 epochs, the OLS score rises to 43.7.

Table 2 shows the performance comparison on OpenLane-V2 $subset\_B$. Our proposed TopoMLP exceeds other models in all metrics when using the same ResNet-50 backbone. Particularly in terms of topology performance, it surpasses TopoNet by a large margin (7.6 *v.s.* 2.5 on $TOP_{ll}$, 17.8 *v.s.* 14.2 on $TOP_{lt}$). Moreover, the performance boost is also observed when integrating more powerful backbones. Overall, these results significantly highlight the efficacy of our TopoMLP model.

### 4.4 ABLATION STUDY

In this section, we study several important components of our method and conduct ablation experiments on OpenLane-V2 $subset\_A$.

**Analysis on Lane Detection.** We investigate the effect of different settings in lane detection. **i)** In Table 3 (a), the improvement in lane detection and lane-lane topology performance is clear when the lane query increases from 200 to 300. However, it is observed that any further increase does not contribute to additional improvement. To balance the model efficiency and performance, the number of lane query is set to 300. **ii)** Table 3 (b) illustrates the influence of a control point in Bézier modeling. Empirically, we choose 4 control points for better performance.

**YOLOv8 Proposal on Traffic Detection.** To study the benefit of using YOLOv8 proposals, we test its effect under two settings: using ResNet-50 and Swin-B backbone. The main results are shown in Table 1, marked by "*". It is well seen that using YOLOv8 predictions as proposal queries consistently improves the detection performance, indicating the effectiveness of YOLOv8 proposals. Moreover, it is worth noticing that TopoMLP without YOLOv8 still achieves higher traffic detection scores than other counterparts.

**Representation Way in Topology Reasoning.** It is also of interest to analyze the impact of different lane and traffic element representation methods in topology reasoning. **i)** We first analyze the lane representation in lane-lane topology, which additionally integrates lane coordinates. As shown in Table 3 (c), removing it leads to a minor performance degradation on $TOP_{ll}$, which indicates that the explicit lane position is useful for topology reasoning. Moreover, abandoning L1 loss for intersection point supervision also causes a score decrease. **ii)** Table 3 (d) explores the impact of incorporating

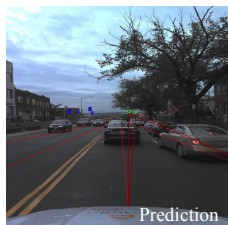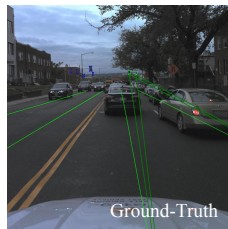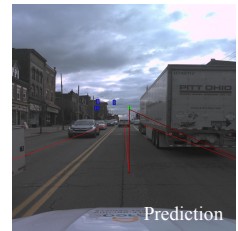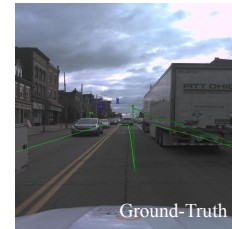

Prediction    Ground-Truth    Prediction    Ground-Truth

Figure 4: **Traffic detection and lane-traffic topology from TopoMLP.** Our method can detect the traffic elements in the front view and associate them with lanes. The green represents GT, while the red means our prediction. Traffic predictions are grounded by different colors in terms of category.

| Lane Queries | $DET_l$ | $DET_t$ | $TOP_{ll}$ | $TOP_{lt}$ | OLS |
|---|---|---|---|---|---|
| 200 | 28.2 | 49.9 | 6.1 | 20.2 | 36.9 |
| 300 | **28.3** | **50.0** | 7.2 | **22.8** | **38.2** |
| 500 | 27.9 | 49.6 | **7.3** | 22.4 | 38.0 |

(a) Different number of lane queries.

| Control Points | $DET_l$ | $DET_t$ | $TOP_{ll}$ | $TOP_{lt}$ | OLS |
|---|---|---|---|---|---|
| 3 | 26.6 | 49.9 | 7.0 | 21.5 | 37.3 |
| 4 | **28.3** | **50.0** | **7.2** | **22.8** | **38.2** |
| 5 | 27.8 | 48.5 | 6.6 | 21.5 | 37.1 |

(b) Different number of control points.

| LL Topo | $DET_l$ | $DET_t$ | $TOP_{ll}$ | $TOP_{lt}$ | OLS |
|---|---|---|---|---|---|
| Ours | **28.3** | 50.0 | **7.2** | **22.8** | **38.2** |
| *w/o* position | 27.9 | **50.9** | 6.9 | 21.6 | 37.9 |
| *w/o* L1 loss | 26.6 | 50.9 | 6.5 | 22.1 | 37.5 |

(c) LL Topo is the short name for lane-lane topology. "*w/o* position" means removing lane coordinate embedding. "*w/o* L1 loss" means removing the supervision of interaction points.

| LT Topo | $DET_l$ | $DET_t$ | $TOP_{ll}$ | $TOP_{lt}$ | OLS |
|---|---|---|---|---|---|
| Ours | 28.3 | **50.0** | **7.2** | **22.8** | **38.2** |
| *w/o* transform | **28.4** | 49.3 | 7.2 | 21.4 | 37.7 |
| *w* only box | 28.2 | 49.6 | 7.1 | 22.0 | 37.8 |

(d) LT Topo is the short name for lane-traffic topology. "transform" means using view transformation matrix on lane feature. "only box" means using bounding box as traffic representation.

Table 3: **The ablation studies** of different components in the proposed TopoMLP. The experiments are conducted on OpenLane-V2 $subset\_A$. We bold the best scores.

the view transformation matrix into the lane feature for lane-traffic topology. It suggests that the integration of this matrix into lane representation improves the reasoning of lane-traffic topology (22.8 *v.s.* 21.4 on $TOP_{lt}$). **iii)** Using the bounding boxes of traffic elements to replace traffic features results in a drop on $TOP_{lt}$ (22.8 *v.s.* 22.0), as shown in the last row of Table 3 (d). This is because only adopting boxes lacks category information. Despite the advantages of position embedding, a single MLP network proves sufficient for achieving high-performance topology reasoning.

## 4.5 VISUALIZATION

We visualize the lane detection and lane-lane topology reasoning results in Fig. 3 by projecting 3D lanes into images. Despite potential challenges like intricate intersections or occluded centerlines, TopoMLP well predicts the centerlines and constructs a lane graph in BEV. Fig. 4 displays the results of traffic detection as well as lane-traffic topology reasoning. As clearly shown, TopoMLP identifies the majority of traffic elements, even small objects, and allocates them to the appropriate lanes.

## 4.6 MORE DISCUSSION

Before stepping into our thorough analysis, let's revisit the definition of the topology metric. Given the ground-truth graph $G = (V, E)$ and the predicted one $\hat{G} = (\hat{V}, \hat{E})$, we establish a projection on the vertices such that $V = \hat{V}' \subseteq \hat{V}$. This projection utilizes the Fréchet and IoU distances to measure similarity among lane centerlines and traffic elements respectively. Within the predicted $\hat{V}'$, we consider two vertices as being connected if the confidence of the edge surpasses 0.5. Subsequently, the TOP score is derived by averaging vertice mAP between $(V, E)$ and $(\hat{V}', \hat{E}')$ overall all vertices:

$$\text{TOP} = \frac{1}{|V|} \sum_{v \in V} \frac{\sum_{\hat{n}' \in \tilde{N}'(v)} P(\hat{n}') \mathbf{1}_{\text{condition}}(\hat{n}' \in N(v))}{|N(v)|}, \tag{7}$$

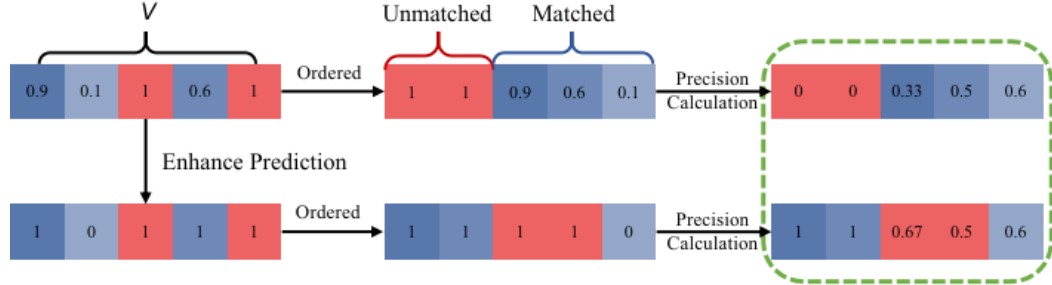

Figure 5: **The illustration of loophole on TOP metric.** Enhancing the prediction scores leads to true positives prior to some false positives from unmatched instances, further improving precision.

| Method | $DET_l$ | $DET_t$ | Original Prediction | | | Enhanced Prediction | | |
| --- | --- | --- | --- | --- | --- | --- | --- | --- |
| | | | $TOP_{ll}$ | $TOP_{lt}$ | OLS | $TOP_{ll}$ | $TOP_{lt}$ | OLS |
| TopoNet (Li et al., 2023) | 27.1 | 44.8 | 4.0 | 20.6 | 34.4 | 11.5(+7.5) | 21.6(+1.0) | 38.1(+3.7) |
| TopoMLP (Ours) | 28.3 | 50.0 | 7.2 | 22.8 | 38.2 | 19.0(+11.8) | 23.4(+0.6) | 42.2(+4.0) |

| Method | $DET_l$ | $DET_t$ | $TOP_{ll}^{\dagger}$ | $TOP_{lt}^{\dagger}$ | $OLS^{\dagger}$ | $TOP_{ll}^{\dagger}$ | $TOP_{lt}^{\dagger}$ | $OLS^{\dagger}$ |
| --- | --- | --- | --- | --- | --- | --- | --- | --- |
| TopoNet (Li et al., 2023) | 27.1 | 44.8 | 2.0 | 20.4 | 32.8 | 1.0 | 20.9 | 32.0 |
| TopoMLP (Ours) | 28.3 | 50.0 | 4.5 | 22.1 | 36.3 | 1.9 | 22.5 | 34.5 |

Table 4: **The comparison on the original TOP metric and our adjusted TOP (marked by †)** when using enhanced prediction or not. TopoNet is reimplemented by ours using the same backbone ResNet-50 with our TopoMLP. The experiments are conducted on OpenLane-V2 $subset\_A$.

where $N(v)$ is the ordered list of neighbors of vertex $v$ ranked by confidence and $P(v)$ is the precision of the $i$-th vertex $v$ in the ordered list.

We provide a toy example of the important loophole in Fig. 5. A crucial point hinges on the precision of the ordered list. For those unmatched instances that our detector cannot identify, their confidence scores are defaulted to 1.0. That is, there are lots of false positives with high confidence. Suppose we push the prediction confidence into 1.0/0.0 in terms of 0.5, our prediction with true positives will have a higher confidence, hence, leading to enhanced precision. The quantitative results are shown in the first two rows of Table 4. Using this strategy to enhance prediction leads to consistent performance improvement, including TopoNet and our method TopoMLP.

To tackle this issue, we suggest a novel TOP metric incorporated with a correctness factor. Let's symbolize the enhanced precision as $P(v)^{\dagger}$. The adjusted TOP metric $TOP^{\dagger}$ is formulated as:

$$\text{TOP}^{\dagger} = \frac{1}{|V|} \sum_{v \in V} \frac{\sum_{\hat{n}' \in \hat{N}'(v)} P(\hat{n}') \mathbf{1}_{\text{condition}} \cdot (\hat{n}' \in N(v)) \frac{N_{TP}}{(N_{TP} + N_{FP})}}{|N(v)|}, \quad (8)$$

where $N_{TP}$ is the number of true positives and $N_{FP}$ is the number of false positives. In the last two rows of Table 4, we evaluate TopoNet and TopoMLP on the adjusted topology metric, demonstrating its capability to effectively shield against the "attack". Despite the alteration in the metric, TopoMLP still surpasses other methods such as TopoNet in performance.

## 5 CONCLUSION

In this paper, we propose a simple yet strong pipeline for driving scene topology, named TopoMLP. It starts a significant observation that the reasoning performance is limited by the detection scores. Therefore, we first focus on designing two powerful detectors for 3D lane detection and 2D traffic detection, respectively. As for topology reasoning, combining the appreciated position embedding and an elegant MLP network is enough to achieve impressive performance. TopoMLP is the 1st solution for 1st OpenLane Topology in Autonomous Driving Challenge. We hope our work opens up new insights into exploring driving topology reasoning.

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
