# OpenReview forum: "TopoMLP: A Simple yet Strong Pipeline for Driving Topology Reasoning"
_ICLR.cc/2024/Conference — ICLR 2024 poster_

### Official Review · Reviewer_Cep6 · 2023-10-23

**Soundness:** 3 good
**Presentation:** 3 good
**Contribution:** 3 good
**Rating:** 8
**Confidence:** 4

**Summary:**

The paper presents a framework called TopoMLP for driving topology reasoning, which aims to accurately predict the relationship between lanes and traffic elements in autonomous driving scenarios. The key idea of TopoMLP is to leverage a simple yet effective approach based on MLP networks for topology reasoning. The authors also address the limitations of the topo metric of OpenLane V2 and provide an improved version. This topology reasoning algorithm achieves the state-of-the-art performance in the lane-lane and lane-traffic element relationship reasoning task on the OpenLane V2 datasets.

**Strengths:**

1. The fundamental idea presented in the paper is solid, and the overall writing is of high quality.
2. The paper provides experimental results that effectively support the proposed method.
3. The authors enhance the query-based detectors by integrating additional reference boxes generated by YOLOv8.
4. The topology reasoning method employed is both simple and effective.

**Weaknesses:**

1. One weakness of the proposed method is its inability to accurately represent complex shapes of map elements such as curbs using bezier curves. This limitation restricts the capability of the method for effective map learning and topology reasoning.
2. Additionally, while the authors have made improvements to the query-based traffic elements detection model, the box proposals generated are still outputs from a yolov8 model, which leaves some aspects of the work incomplete.

**Questions:**

1. Regarding the scenarios depicted in Figure 3, where the intersection area on both the left and right sides is either invisible or not directly observable, the displayed results appear to be very promising. I would like to confirm whether these results were obtained by overfitting the dataset.

---

> ### Author Response · Authors · 2023-11-21
> **The Response to Reviewer Cep6**
>
> We thank reviewer Cep6 for the valuable time and constructive feedback.
>
> ---
>
> **Q1:  About the Bezier representation.**
>
> **A1:** Thanks for your comment. While it's true that Bezier curve may not perfectly represent the complexity of map elements, its has great potential for delivering superior results through piecewise Bezier curves. We'd like to draw your attention to a pivotal study titled "End-to-end vectorized HD-map construction with piecewise Bezier curve" (CVPR2023).  This research leverages piecewise bezier curve to obtain the state-of-the-art score in HD-map perception task. In addition, our method can be extended into other lane representation, such as the one in MapTR (Maptr: Structured modeling and learning for online vectorized hd map construction, ICLR2022). Maybe it can serve as the exploration of our future work.
>
> ---
>
> **Q2:  About YOLOv8 outputs.**
>
> **A2:** Good comment! We make detailed clarification below.
>
> **Motivation of using YOLOv8:** Our framework for traffic detection is fully DETR-based. YOLOv8 is optional for the query initialization of our framework. The query initialization can also be randomized. Both solutions are not competitors yet different choices.
>
> **Why YOLOv8 performs better:** Randomly initialized query-based detector underperforms against using specific detector YOLOv8 as query initialization. We think this result is reasonable. This is due to the vast number of augmentation techniques, like mixup and mosaic aug when training YOLOv8. However, it is challenging to integrate these augmentation techniques into our end-to-end framework, which may hinder the performance of 3D lane detection.
>
> ---
>
> **Q3: About visualization results.**
>
> **A3:** The results displayed in Fig.3 of our paper are not overfitting the dataset. Our model is able to predict that there are lanes in some invisible areas, even though the estimation of the lane quantity and length might not be perfect (see comparison between lane prediction and ground-truth in this [link](https://www.dropbox.com/scl/fo/a0m3aow7m747tmo8mdwvm/h?rlkey=t7mlzvfsxim582i453y4hah30&dl=0)).
>
> ---
>
> We appreciate your thoughtful review and we hope to address your concerns. Please let us know if you'd like any further discussion.

---

> > ### Comment · Reviewer_Cep6 · 2023-11-23
> >
> > Thanks for the replies. My concerns have been solved and  I will keep my score.

---

### Official Review · Reviewer_bKsZ · 2023-10-31

**Soundness:** 3 good
**Presentation:** 3 good
**Contribution:** 2 fair
**Rating:** 6
**Confidence:** 5

**Summary:**

The paper presents a methodology named TopoMLP for topology reasoning in autonomous driving. Topology reasoning is essential for understanding road scenes and presenting drivable routes, requiring the detection of lane centerlines and traffic elements and reasoning their topology relationships, such as lane-lane and lane-traffic element topologies. The authors introduce a powerful 3D lane detector and an improved 2D traffic element detector, emphasizing the significance of detection performance on lane and traffic elements in influencing topology scores. They develop two simple MLP-based heads for topology generation, based on impressive detection performance. The proposed model, TopoMLP, achieves state-of-the-art performance on the OpenLane-V2 benchmark and ranks 1st in the 1st OpenLane Topology in Autonomous Driving Challenge. The authors also identify a loophole in the current topology metrics, and propose a correctness factor to enhance it.

**Strengths:**

1. The authors conduct a comprehensive investigation into the significance of detection performance in topology reasoning, utilizing the OpenLane-V2 metrics. They highlight that with a detection score of 28.3, the topology score can only reach a maximum of 7.5, even if the topology reasoning results are identical to the ground truth.
2. In the above scenario, their model achieves a topology score of 7.2. This demonstrates the model's superior performance in topology reasoning compared to other models evaluated using the OpenLane-V2 metrics.
3. They propose using lane coordinate embedding and extra-supervision for interaction points to increasing LL topology reasoning performance. The LT topology can also benefited from view transformation embbeding and traffic element features.
4. They have established a strong baseline framework for topology reasoning for autonomous driving, and plan to make it open-source, laying a solid foundation for subsequent work. They also pointed out a flaw in an existing metric, which has contributed to the advancement and refinement of this field.

**Weaknesses:**

1. Although the overall performance of TopoMLP is impressive, it primarily stems from the overall framework and training strategy, where the novelty being somewhat limited.
2. The several modules proposed in the paper to enhance performance are effective, but they are quite straightforward and intuitive, lacking more refined and intricate design.
3. Some parts of the manuscript are unclear and could be further improved; errors and suggestions for improvement are detailed later in the Questions.

**Questions:**

1. TopoMLP has a higher lane-lane topology score (7.2) compared to the previous state-of-the-art TopoNet (4.1) under the similar detection score. However, even without your specific designs, the performance of TopoMLP remains strong in the ablation study Table 3(c) (6.9 and 6.5). Can you explain the main reason for the strong TOP_ll performance of your model compared to TopoNet?
2. Although the introduction of YOLOv8 can significantly improve performance, why not directly use YOLOv8's features for LT relationship inference, but use its detection results as proposals of a DETR?
3. Can you clarify why the "adjusted TOP" dropped in the "enhanced prediction"? Do the true positives and false positives in N_TP and N_FP refer to the matching in detection or topology?
4. Some errors in manuscript:
   - In Figure 2, the transformation matrix A is added to the Traffic Features. However, in Section 3.4, it is mentioned that it is "summed with the lane query embedding."
   - In Page 6, Section 4.2, the subscript of "The loss coefficient of lane-traffic topology loss" should be "lt" instead of "ll".

---

> ### Author Response · Authors · 2023-11-21
> **The Response to Reviewer bKsZ**
>
> We thank reviewer bKsZ for the valuable time and constructive feedback.
>
> ---
>
> **Q1: Somewhat limited novelty.**
>
> **A1:** Thanks for your comment. In this work, we mainly contribute novel insights instead of a novel framework. After a set of in-depth analysis, we propose a “first-detect-then-reason” philosophy for better topology prediction. As the first step of topology prediction, detection has a greater impact on overall driving topology reasoning.
>
> Moreover, we would like to claim that complicated topology head is not necessary in this task. Please refer to the Table responding to Reviewer tnze.  Our simple MLP-based topology head (with SwinB backbone) can reach 95%(9.5/10.0) performance compared to using topology ground-truth, as shown in Fig.1 of our paper. An intuitive and elegant method like this can indeed be a good thing for the new emerging area.
>
> Overall, it's our hope that our discussions and methodology can offer fresh perspectives towards shaping the future of driving topology reasoning.
>
> ---
>
> **Q2: The main reason for strong TOP_ll performance.**
>
> **A2:** Good comment! The superior TOP_ll score of our model is attributed to the fact that TopoMLP has a higher detection recall. Through our statistics, TopoNet yields a recall of 0.36, while TopoMLP achieves a recall of 0.54. This increased recall boosts the number of matched lanes, which in turn leads to improved topology scores.
>
> ---
>
> **Q3: Using YOLOv8’s features.**
>
> **A3:** On the one hand, we use proposals from YOLOv8 as the query initialization of DETR-like framework.  It ensures the unity and elegance of our framework. On the other hand, directly using the object feature from YOLOv8 lacks detailed position information, which is very important for traffic-lane topology.
>
> ---
>
> **Q4: Clarification on adjusted TOP.**
>
> **A4:** Sorry for the confusion. ‘Enhanced prediction’ refers to pushing the prediction confidence to attack the official TOP metric, while our proposed ‘adjusted TOP’ is used to avoid this problem.The adjusted TOP introduces a correctness term (N_TP/(N_TP + N_FP)), which may be a strong penalty. The challenge organizer of OpenLane-V2 has took our advice and modified the TOP metric.  The true positives and false positives refer to the matching in topology.
>
> ---
>
> **Q5: About typo.**
>
> **A5:** Thank you so much for your careful review! The transformation matrix A is summed with the lane query embedding. We will correct the figure and other typos.
>
> ---
>
> We appreciate your thoughtful review and we hope to address your concerns. Please let us know if you'd like any further discussion.

---

> > ### Comment · Reviewer_bKsZ · 2023-11-21
> >
> > Thank you for your response and the additional insights from other reviewers. My questions have been resolved. I will keep my current score.

---

### Official Review · Reviewer_tnze · 2023-10-31

**Soundness:** 3 good
**Presentation:** 3 good
**Contribution:** 3 good
**Rating:** 6
**Confidence:** 4

**Summary:**

The paper proposes a simple and strong method called TopoMLP, on the openlane-v2 benchmark for topology reasoning in the driving scenarios. Authors claim that the key insight is to improve the detection part in order to secure a steady improvement of the topology score. This is also the winner solution for the Openlane-v2 challenge.

**Strengths:**

In general the paper carries some good merits, including:

+ the paper is easy to follow. The contributions are very clear: good detection increases the topology reasoning. Based on the detection module, using two simple MLP heads for topology generation could win among other approaches.

+ this work derives from the participation of this year's Autonmous Driving Challenge. This is the first entry solution, as shown in Table 1 and Table 2. Authors find a problematic on the evalution metric of OLS score. Figure 5 is very illustrative. Using the rectified evaluation, the loophole could be remedied.

**Weaknesses:**

The paper has some obvious shortcomings as follows.

- Motivation / technical novelty seems to be limitted. Following the second paragrah in the introduction, it is empirically found that topology performance is improved with stronger detection. Any insight behind this? There seems not too many discussions on it.

- Some ablations and experiments need to be discussed.

Overall this work falls into the borderline category without too many novelty or insight. However, given (a) the topology reasoning for lane detection is an emerging topic and there are few work in this direction, (b) the paper achieves the best result on the leaderboard of the Challenge, promised reproducible research to the community, the reviewer would slightly lean to encourage the manuscript to be polished in great extent.

**Questions:**

1. I see the pipeline resembles to some extent with TopoNet method, where two branches are capable of recognizing the traffic lights and lane topoploy respectively. Using two simple MLP for the head is the core contribution I suppose. Is there any ablations to verify that using some complicated design (like a Transformer head) could "downgrade" the OLS score?

2. Section 4.6 is a good contribution to the open-sourced community. Do authors communicate with organizers of the Challenge to reconsider the metric and maybe the leaderboard would be refreshed? On further consideration, the current OLS is a general metric. Is it good to design a separate metric to evaluate the topology performance at intersections (which is crucial I suppose)?

3. This work generates the centerline of the lane, which is not the physical representation of the road scene. Is it a good formulation? On a higher level, I think the current form of Openlane-v2 tackles only partial challenges in the static road recognition. How to utilize the topoloy to faciliate downstream task? This could potentially trigger more future research in related domains.

----
Minor

1. Topo in Figure 2. "Topoogy"
2. G^LL in Equation 3 is the operation of Topology in Figure 2? I suggest to align the concept with equation and the figure.
3. How to balance the different weights in Equation 5 of different branches? Are there some ablations or just equally distributed?

---

> ### Author Response · Authors · 2023-11-21
> **The Response to Reviewer tnze**
>
> We thank reviewer tnze for the valuable time and constructive feedback.
>
> ---
>
> **Q1: About our motivation.**
>
> **A1:** Thanks for your comment.  In this work, we would like to demonstrate that a complicated topology head is not that necessary for driving topology reasoning. Instead, the design of high-performance detectors for lane and traffic element plays a more important role. Our MLP-based topology head, backed by ResNet50, has achieved 96% (7.2/7.5) performance relative to using topology ground-truth, based on Fig.1 in our paper.  Therefore, optimizing topology reasoning performance hinges on creating superior detectors, which presents a bottleneck in the process.
>
> Considering the examples given in this [link](https://www.dropbox.com/scl/fo/a0m3aow7m747tmo8mdwvm/h?rlkey=t7mlzvfsxim582i453y4hah30&dl=0), inaccurate detection (Fig.(a)) result in an unsatisfactory assessment of the topology, despite the seemingly advanced lane topology connection (Fig.(b)). Specifically, the evaluation of topology first involves the alignment of prediction and ground-truth lanes. Following this, the overlapped lanes are utilized for the extraction of predicted topology values, which enable the calculation of the graph-based mAP. If the lanes are undetected or not matched with the ground-truth lanes according to the Frechet distance (with varying thresholds), these topology values will be classified as negatives.
>
> In summary, the basic detection is of great importance for the following topology reasoning, in such “first-detect-then-reason” pipeline. We hope these discussions will bring fresh perspectives to the developing field of driving topology reasoning.
>
> ---
>
> **Q2: Evaluation of other modules in topology head.**
>
> **A2:** Good comment! Following your suggestion, we evaluate two complicated topology heads, i.e.,  Transformer-based head and graph-based head. As shown in below table, our MLP-based head has similar performance with these two methods. It supports our opinion that a complicated topology head is not that necessary for driving topology reasoning.
>
> | Topology Head    | DET_l | DET_t | TOP_ll | TOP_lt | OLS  |
> | ------------------ | ------- | ------- | -------- | -------- | ------ |
> | Transfomer-based | 28.0  | 50.0  | 7.0    | 22.5   | 37.9 |
> | Graph-based      | 28.2  | 49.2  | 7.2    | 22.0   | 37.7 |
> | TopoMLP          | 28.3  | 50.0  | 7.2    | 22.8   | 38.2 |
>
> ---
>
> **Q3: Discussion with organizers of the Challenge.**
>
> **A3:** Yes, we have discussed this problem with organizers of the Challenge. They have took our advice and decided to modify the TOP metric. The adjusted TOP metric will be used in the next competition. As for the separate metric on evaluating topology at interactions, it is indeed sigificiant and we will provide your feedback to discuss with challenge organizers.
>
> ---
>
> **Q4: About the representation of centerline.**
>
> **A4:** Good comment! While the centerline of the lane is not the physical representation of the road scene, but it symbolizes a higher level idea, namely, the driving route. First,  the centerline at interactions can be formulated, defining potential routes at interactions or road convergences. Second, it can integrate traffic element information, providing a comprehensive perception of the road. Last but not least, the centerline of Openlane-v2 offers driving routes, which bring many benefits for downstream task that requires potential routes, such as motion prediction and ego-car planning. Overall, the common lane detection can be viewed as a parallel task with the driving topology reasoning. Both will promote the development of autonomous driving.
>
> ---
>
> **Q5: Minor Questions.**
>
> **A5:** Thanks for your careful review. We will correct the typos and alignment. As for the hyparameters in Eq.5, we make some ablations for determining the values. The below table lists the results on different regression loss weights of lane detection head.
>
> | Lane reg weight | DET_l | DET_t | TOP_ll | TOP_lt | OLS  |
> | ----------------- | ------- | ------- | -------- | -------- | ------ |
> | 0.0075          | 26.6  | 49.8  | 7.0    | 21.4   | 37.3 |
> | 0.01            | 27.9  | 49.6  | 7.1    | 22.4   | 37.9 |
> | 0.02 (Ours)     | **28.3** | **50.0**  | **7.2**    | **22.8**   | **38.2** |
> | 0.05            | 28.2  | 49.6  | 7.0    | 22.0   | 37.8 |
>
> ---
>
> We appreciate your thoughtful review and we hope to address your concerns. Please let us know if you'd like any further discussion.

---

> > ### Comment · Reviewer_tnze · 2023-11-21
> >
> > Thanks for the clarification! I really appreciate it. After looking through the rebuttal and other comments from fellow reviewers, I’d like to keep the rating as weak acceptance.

---

### Official Review · Reviewer_MJLZ · 2023-11-01

**Soundness:** 3 good
**Presentation:** 3 good
**Contribution:** 3 good
**Rating:** 6
**Confidence:** 3

**Summary:**

The authors introduces TopoMLP driving topology reasoning using vehicle-mounted sensors.
The model starts with two strong decoder to detect the lanes and the traffic elements.
For the lanes, the model uses Bezier curves as lane queries for 3D centerline position embedding.
For the traffic elements, it uses proposals from YOLOv8 and the corresponding multi-scale features, and feed them to a series of deformable decoder layers.

These information then are passed to two MLP network reasoners to perform the final lane-lane and lane-traffic-element topology reasoning.

Both of these modules apply pair-wise representations for their respective inputs.
However, Lane-Traffic reasoning also takes in camera intrinsic and extrinsic information.

The loss functions uses focal loss throughout, for classification for both lane and traffic detection, and lane-lane and lane-traffic topology.
In addition, lane detection loss also includes L1 loss for lane regression and lane-lane topology, while traffic detection loss also includes GIoU loss for bounding box regression.

TopoMLP is tested on OpenLane-V2 benchmark, 2000 scenes in total, against a number of STOA techniques.
The model is able to surpass the rest of the techniques, by showing a much better topological reasoning capability, while achieving a decent detection accuracy.
The ablation study shows that the value of using YOLOv8 object detection, explicit lane coordinates, and injecting transformation matrix.
Finally, the paper proposes an enhanced TOP score incorporating correctness factor, which allows for more accurate precision.

**Strengths:**

The proposed model improves the STOA on OpenLane dataset by refining its various components.
The proposed enhanced TOP score also shed lights to a more accurate evaluation.

**Weaknesses:**

It is hard to make sense of why the TOP numbers are quite low, especially lane-lane topology.
It would be good to add what is an acceptable number for robust driving.
There is not a lot of discussion of what is still not solvable, to allow readers to understand the shortcomings of the methods.
This would add discussions on what is the next steps to take.

**Questions:**

Would you be able to ground the TOP numbers with what is required for autonomous driving.
Also discussions on what is not yet solvable.

Typos, Grammar, etc.:
pg. 4, figure 2 caption: “Topoogy” —> “Topology”

**Details Of Ethics Concerns:**

None.

---

> ### Author Response · Authors · 2023-11-21
> **The Response to Reviewer MJLZ**
>
> We thank reviewer MJLZ for the valuable time and constructive feedback.
>
> ---
>
> **Q1: About TOP_ll score.**
>
> **A1:** Thanks for your careful review. We make detailed clarification below.
>
> **Why TOP numbers are low:** TOP_ll score is heavily limited by the lane detection performance. In specific, the topology evaluation first matches the predicted lanes and ground-truth lanes, and then bases the matched lanes to extract the predicted topology values for calculating graph-based mAP. If the lanes are not detected or matched with ground-truth in terms of Frechet distance (with different thresholds), the topology values will be regarded as negatives.  By the way, from Fig.1 in our manuscript, our TopoMLP with ResNet50 has achieved 96%(7.2/7.5) topology scores compared to using ground-truth topology. In a word, the topology performance depends heavily on detection performance.
>
> **Discussion on acceptable number:** Although some TOP_ll scores are low, their corresponding visualization performance is promising. For example, the sample from Fig.3 in our manuscript has only TOP_ll=2.8, while the topology connection across lanes looks acceptable (the thin lines are topology connections). This is because detection metric is strict, further leading to many unmatched lanes for topology evaluation and decreasing the topology scores. See the comparison between the lane prediction and ground-truth in this [link](https://www.dropbox.com/scl/fo/a0m3aow7m747tmo8mdwvm/h?rlkey=t7mlzvfsxim582i453y4hah30&dl=0).
>
> **Possible solutions:** 1) Our research findings show that the effectiveness of detectors has a significant impact on the overall topology performance.  To boost basic detection performance, it requires additional time and effort. 2) It is still an open problem for feature alignment between 2D traffic element and 3D lane in a unified framework. 3) Utilizing a large language model could be a potential solution to address this driving reasoning.
>
> ---
>
> **Q2: About typos.**
>
> **A2:** Thanks for pointing out this typo, and we will correct it in our revision.
>
> ---
>
> We appreciate your thoughtful review and we hope to address your concerns. Please let us know if you'd like any further discussion.

---

> > ### Comment · Reviewer_MJLZ · 2023-11-22
> >
> > Dear authors,
> >
> > Thank you so much for the replies. After reading them, as well as other discussions, I have decided to keep my score.

---

### Meta-Review · Area_Chair_MsyZ · 2023-12-09

**Metareview:**

The meta-reviewer has carefully read the paper, reviews, rebuttals, and discussions between authors and reviewers. The meta-reviewer agrees with the reviewers that this is a solid submission to ICLR. The paper proposed TopoMLP, for analyzing road topology using vehicle sensors, detecting lanes with Bezier curves and traffic elements with YOLOv8 proposals followed by deformable decoders. It employs two MLP networks to reason about lane and traffic element relationships, incorporating pairwise representations and camera data for lane-traffic analysis. The model uses focal loss for classification tasks, L1 loss for lane regression, and GIoU loss for traffic element bounding box regression. When evaluated on the OpenLane-V2 benchmark, TopoMLP outperformed existing techniques in topological reasoning and detection accuracy. The paper also proposes an enhanced TOP score for improved precision assessment. The meta-reviewer agrees with the reviewers that this paper has a clear contribution to the autonomous driving community and thus recommends acceptance.

**Justification For Why Not Higher Score:**

N/A

**Justification For Why Not Lower Score:**

N/A

---

### Decision · Program_Chairs · 2024-01-16

Accept (poster)